# Enhancing Zeroth-Order Fine-Tuning for LLMs via Gradient-Guided Subspace Selection

## Abstract

As a promising memory-efficient technique, zeroth-order (ZO) optimization enables large language models (LLMs) to bypass costly backpropagation during fine-tuning by estimating gradients through function evaluations. However, to minimize approximate variance in high-dimensional parameter spaces, existing ZO methods focus on exploring the estimate of gradients within random subspaces, neglecting the benefits of searching for more accurate subspaces of LLMs on gradient estimates. Due to inaccurate gradient estimates obtained from random spaces, fine-tuning performance is inevitably degraded, thus compromising the performance of downstream tasks. To address the limitation of existing ZO methods, this paper proposes a novel ZO subspace fine-tuning method named *SVD-0*. Based on singular value decomposition (SVD), SVD-0 can effectively obtain more accurate subspace projection matrices, which can be used to improve the accuracy of gradient estimates. Experimental results on various complex language modeling tasks show that SVD-0 achieves better fine-tuning performance than state-of-the-art ZO methods.

## 1 Introduction

Due to the powerful capabilities of language understanding and reasoning, large language models (LLMs) have demonstrated significant performance on a wide range of tasks, such as mathematical reasoning[15], creative writing [36]. Currently, fine-tuning (FT) the pre-trained foundation model to adapt to downstream tasks has become the mainstream paradigm for AI application development. However, because of the extremely large number of model parameters, traditional first-order (FO) optimization-based fine-tuning methods face the serious challenge of excessive memory consumption. Typically, since the backpropagation process in FO needs to store the activations and optimizer states, the memory requirements of FT are significantly larger than those of reasoning, which seriously limits the development of LLM-based applications.

To achieve memory-efficient FT, existing methods can be classified into two categories, i.e., parameter-efficient fine-tuning (PEFT) methods [25, 16] and zeroth-order (ZO) optimization methods [29]. PEFT methods attempt to reduce the number of trainable parameters to alleviate memory requirements. However, since PEFT methods are still based on FO optimization, they have to consume a lot of memory to store intermediate training results, which severely limits the choice of trainable parameters. ZO optimization methods [29] emerge as a promising alternative by estimating gradients through forward-pass perturbations, eliminating backpropagation's memory overhead. However, conventional ZO methods face a critical challenge: the high variance of gradient approximations in billion-parameter spaces severely degrades optimization efficiency and model performance.

Recent advances in ZO optimization for LLMs, such as SubZero [43] and LOZO [5], attempt to mitigate this issue by constraining perturbations to random low-dimensional subspaces. These

methods are based on the finding that gradient matrices become low-rank during LLM training and fine-tuning [47]. While these subspace methods reduce approximation variance, they fundamentally rely on arbitrary projection matrices that fail to match the low-rank structure implied by the gradient. This limitation stems from a fundamental disconnect - the subspace construction process ignores critical gradient information that could guide more effective parameter updates. Therefore, *how to determine the optimal low-dimensional subspaces without relying on first-order optimizers* poses a fundamental challenge.

The similarity between the gradient estimated by ZO optimizers and the true gradient has been experimentally demonstrated [29]. Consequently, we consider it viable to derive the low-rank structure of the true gradient from the estimated gradient. Based on this idea, we performed a prestudy. The experimental findings indicate that there is considerable similarity between the estimated gradient and the true gradient when comparing their singular value vectors. Therefore, we conclude that applying the SVD decomposition to the gradient estimated by the ZO optimizer enables us to obtain a low-rank structure that closely resembles the true gradient's low-rank structure.

Based on the above motivation, we propose SVD-0, a novel gradient-guided subspace optimization framework that synergizes zeroth-order efficiency with principled subspace discovery. Our key insight is that, while exact first-order gradients remain inaccessible due to memory constraints, ZO gradient estimates contain sufficient directional information to reconstruct high-fidelity subspaces. Specifically, SVD-0 periodically performs singular value decomposition (SVD) on ZO gradient estimates to derive layer-wise projection matrices that capture dominant optimization directions. By preserving the intrinsic structure of the subspace, our method effectively enhances the performance of subspace-based ZO methods. The contributions of this work are summarized as follows:

- We propose a novel method for exploring more accurate subspace projection matrices and conducting layerwise perturbations on low-rank matrices. With periodic updates of the projection matrices, our method continuously captures the subspaces of the varying parameters.

- To overcome the paradox that obtaining subspace projection matrices requires FO gradients, we develop a novel gradient-guided ZO method to approximate these two projection matrices, ensuring low memory usage throughout the entire fine-tuning process.

- We conduct comprehensive experiments on various model scales and language modeling tasks. The corresponding results show the superiority of our method compared to various ZO optimization methods tailored for LLM fine-tuning.

## 2 Related Work

**Memory efficient fine-tuning for LLMs.** Recent work has concentrated on exploring memory-efficient fine-tuning methods to enable LLM fine-tuning on memory-intensive hardware. A critical line of research centers on Parameter-Efficient Fine-Tuning (PEFT) methods [25, 16] by freezing the backbone of LLMs while only tuning a small group of parameters. For instance, LoRA [18] only updates parameters based on low-rank structures while being competitive with full-parameter fine-tuning. LISA [31] distinguishes trainable layers based on their contribution to task-specific performance and freezes other layers to reduce the memory footprint. Further, parameter quantization [24, 12] has played a pivotal role in enhancing memory efficiency. By discretizing model parameters (e.g., from 32-bit to 8-bit or lower precision), quantization methods such as QLoRA [10] and LLM.int8() [9] reduce storage requirements without significant degradation in task performance. Complementary to PEFT and the quantization method, subspace projection techniques have emerged as a powerful strategy to reduce the dimensionality of the optimization space. Galora [47] and FLORA [17] both leverage the low-rank property of gradients to constrain updates on a compact subspace of the full parameter space [19]. By discovering the projection matrices of low-rank subspaces, the memory costs for storing gradients and optimizer states (e.g., the first and second order states in Adam optimizer [21]) are greatly reduced.

**Zero-order optimization.** ZO approaches enable backpropagation-free optimization by approximating exact gradients through finite differences. This flexibility has driven interest in ZO for solving a range of machine learning problems, including on-chip learning, black-box adversarial strategies and memory-efficient LLMs fine-tuning [29, 46]. Despite these strengths, the practical

use of ZO is mainly confined to smaller-scale tasks and models. A critical limitation stems from the high error in its gradient approximations [32], which becomes more pronounced as problems grow larger and more complex, making scaling challenging. To address this issue, approaches such as MeZO-SVRG [13] and DiZO [39] utilize variance-reduction methodologies [28] to mitigate gradient divergence. Furthermore, methods including SparseMezo [27], TeZO [38], and AdaZeta [42] have been proposed to diminish approximation errors by reducing dependence on the parameter dimension through parameter sparsification and tenorization. Subspace methods [30], including SubZero [43] and LOZO [5], are explored to leverage low-rank structures for decreasing the error. Although they effectively alleviate the variance of gradient approximation, the randomly generated projection matrices cannot precisely reflect the transformation between the subspace and full space, leading to model performance degradation.

## 3 Prestudy

In exploring the alignment between estimated ZO and true FO gradients in the parameter spaces of large language models, we perform a targeted analysis using the OPT-1.3B model [45] on the RTE task [7, 1, 14, 2]. For every 50 training steps, we determine the exact FO gradients through backpropagation with a batch size of 16 and ZO gradient estimates via MeZO's simultaneous perturbation method [29]. Subsequently, we apply singular value decomposition (SVD) to both gradient matrices. We then assess the cosine similarity between the singular value vectors.

Figure 3 shows that the singular vectors have a high cosine similarity, indicating that the ZO gradients maintain key optimization directions and show a similar low-rank structure. This observation supports our main hypothesis. ZO gradient estimates possess enough spectral information to reconstruct FO-guided low-rank subspaces. The preserved directional accuracy suggests that limiting ZO perturbations to the primary gradient subspaces could reduce approximation variance while still maintaining effective updates. These concepts lay the groundwork for our SVD-0 optimization framework, which systematically uses the inherent structure in the ZO gradient estimates to achieve FO-guided efficiency without the computational burden of back-propagation.

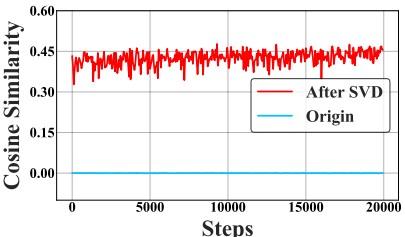

Figure 1: Cosine similarity of estimated ZO gradients compared to true gradients (Original and After SVD).

## 4 Methodology

### 4.1 Overview of Our Method

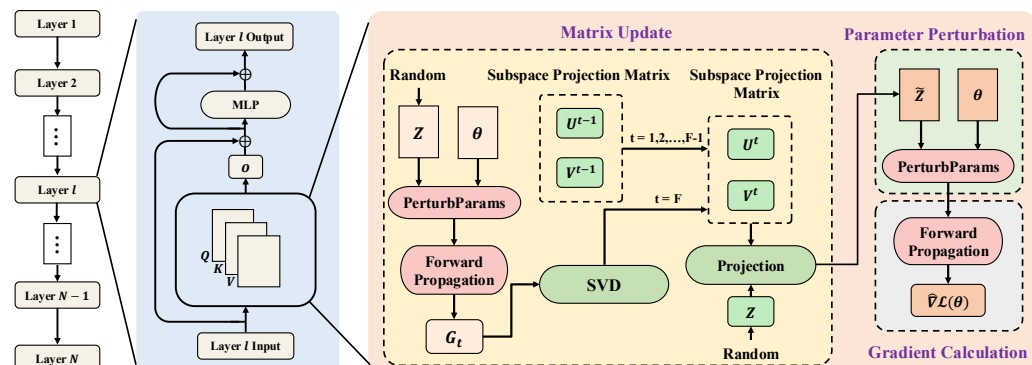

Figure 2: Framework and workflow of our SVD-0 method.

Figure 2 illustrates our approach, centering on two primary components, i.e., the matrix update module and the parameter perturbation module. The matrix update module deals with the computations and adjustments of projection matrices, denoted by $U \in \mathbb{R}^{m \times r}$ and $V \in \mathbb{R}^{n \times r}$. These matrices, in

conjunction with a low-dimensional random matrix $\boldsymbol{Z} \in \mathbb{R}^{r \times r}$, are utilized to produce a low-rank perturbation $\tilde{\boldsymbol{Z}}$.

Within the first module (i.e., the matrix update module), we introduce an innovative and precise approach to acquire the matrices $U$ and $V$, as detailed in Algorithm 1. Traditional approaches often utilized random low-rank perturbation matrices [5, 43]. This randomness contributed to uncertainty in the gradient update process during training. In contrast, our approach computes the $U$ and $V$ matrices based on the gradient information derived using the MeZO method [29] before each update.

---

**Algorithm 1** GenerateProjMatrix($\boldsymbol{G}, r$)

**Input**: i) $\boldsymbol{G}$, estimated gradient of parameter matrix; ii) $r$, rank.
**Output**: $\boldsymbol{U}, \boldsymbol{V}$, projection matrices.

1: $(\boldsymbol{P}, \boldsymbol{S}, \boldsymbol{Q}) \leftarrow \text{SVD}(\boldsymbol{G})$
2: $\boldsymbol{U} \leftarrow \boldsymbol{P}[:, :r]$
3: $\boldsymbol{V} \leftarrow \boldsymbol{Q}[:, :r]$
4: **return** $\boldsymbol{U}, \boldsymbol{V}$

---

The second module serves to perturb the parameters, as described in Algorithm 2. Common enhancements, like SubZero [43] and the SVD-0 approach suggested here, reformulate the update mechanism by adopting a low-rank perturbation method. As illustrated in Figure 2, the low-rank perturbation $\tilde{\boldsymbol{Z}} \in \mathbb{R}^{m \times n}$ is determined in the following manner:

$$\tilde{\boldsymbol{Z}} = \boldsymbol{U}\boldsymbol{Z}\boldsymbol{V}^T, \tag{1}$$

where $\boldsymbol{Z} \in \mathbb{R}^{r \times r}$ is a random perturbation matrix sampled from $N(0, 1)$. Consequently, the

---

**Algorithm 2** PerturbParams($\boldsymbol{W}, \mathcal{U}, \mathcal{V}, r, \varepsilon, s$)

**Input**: i) $\boldsymbol{W}$, model parameter set; ii) $\mathcal{U}$ and $\mathcal{V}$, projection matrix sets; iii) $r$, rank; iv) $\varepsilon$, perturbation scale; v) $s$, seed.
**Output**: Model parameter set after perturbation.

1: ResetGenerator($s$)
2: **for** $i = 1, 2, \ldots, l$ **do**
3: $\quad \boldsymbol{Z}_i \leftarrow \text{GeneratePerturbMatrix}(r)$
4: $\quad \boldsymbol{W}_i \leftarrow \boldsymbol{W}_i + \varepsilon \boldsymbol{U}_i \boldsymbol{Z}_i \boldsymbol{V}_i^T$
5: **end for**
6: **return** $\boldsymbol{W}$

---

parameter $\theta_t \in \mathbb{R}^{m \times n}$ during the $t^{th}$ iteration is determined by $\theta_t^{\pm} = \theta \pm \tilde{\boldsymbol{Z}} = \theta \pm \boldsymbol{U}\boldsymbol{Z}\boldsymbol{V}^T$. Thus, the gradient is approximated using two forward evaluations as expressed below:

$$\widehat{\nabla}\mathcal{L}(\theta_t^{\pm}) = \frac{\mathcal{L}(\theta_t^+; \mathcal{B}) - \mathcal{L}(\theta_t^-; \mathcal{B})}{2\epsilon} \boldsymbol{U}\boldsymbol{Z}\boldsymbol{V}^T. \tag{2}$$

## 4.2 Gradient-Guided Subspace Projection Matrix Acquisition

Existing approaches to projection matrix construction consist of a spectrum of techniques, ranging from randomized sampling methods [5, 43] to computationally intensive deterministic algorithms [48]. Although the former is computationally efficient, it has the defect of insufficient approximation accuracy due to randomness. The latter introduces significant computational overhead while not significantly improving the approximation accuracy. To address this limitation, we propose a balancing strategy based on adaptive subspace decomposition, as shown in lines 4-7 of Algorithm 3.

To retain the advantage of memory efficiency of zero-order optimizations, we calculate the gradient using the MeZO [29] method, as shown in lines 5-6 of Algorithm 3. Before calculating the projection matrix each time, the gradient calculation is required. Then, as shown in the algorithm 1, the $U$ and $V$ matrices are updated according to the gradient obtained this time. We use the SVD method to calculate the projection matrix. Through this method, the original gradient is projected onto a compact space $\boldsymbol{R} \in \mathbb{R}^{r \times r}$: $\boldsymbol{R} = \boldsymbol{U}^T \boldsymbol{G} \boldsymbol{V}$. After that, we can generate a low-rank perturbation $\boldsymbol{Z}$ in this space, as shown in lines 3-4 of the Algorithm 2, and then use the previously calculated $U$ and $V$ matrices to restore this low-rank perturbation to the original high-rank space. In this way, we can successfully apply gradient-based low-rank perturbations to the parameters, and this process will not introduce additional overhead compared to the traditional ZO method (i.e., MeZO).

## 4.3 Periodical Subspace Update

As mentioned above, we obtain the gradient using the MeZO [29] method and obtain the projection matrices $\boldsymbol{U}$ and $\boldsymbol{V}$ by SVD. These two projection matrices jointly determine the gradient approximation and the parameter update of the $t^{th}$ step. However, this iterative update method presents a critical trade-off between computational efficiency and subspace adaptability. High-frequency updates restrict the complete evolution of the gradient subspace while incurring substantial computational costs, particularly due to the need for gradient recomputation prior to each projection matrix update.

**Algorithm 3** SVD-0

---

**Input**: i) $\boldsymbol{W}_i \in \mathbb{R}^{m_i \times n_i}, i = 1, \dots, l$, parameter matrix in the $i$-th layer; ii) $\mathcal{L}$, loss; iii) $T$, step budget; iv) $\epsilon$, perturbation scale; v) $\{\eta^t\}$, learning rate schedule; vi) $F$, subspace update frequency; vii) $r$, rank.

1: **for** $t = 1, \dots, T$ in parallel **do**
2:     $\mathcal{B}^t \leftarrow$ SampleMinbatch $(s^t)$ {Sample a minbatch $\mathcal{B}^t \subset \mathcal{D}$ and a random seed $s^t$}
3:     **for** $i = 1, 2, \dots, l$ **do**
4:       **if** $t \mod F \equiv 0$ **then**
5:         $G_i \leftarrow$ EstimateGradient$(W_i^t, \epsilon)$ {Estimate the gradient of $W_i^t$ using the same way of MeZO}
6:         $\boldsymbol{U}_i^t, \boldsymbol{V}_i^t \leftarrow$ GenerateProjMatrix$(G_i, r)$
7:       **else**
8:         $\boldsymbol{U}_i^t \leftarrow \boldsymbol{U}_i^{t-1}, \boldsymbol{V}_i^t \leftarrow \boldsymbol{V}_i^{t-1}$
9:       **end if**
10:     **end for**
      $\{\boldsymbol{W}^t = \{\boldsymbol{W}_i^t\}_{i=1}^l, \mathcal{U}^t = \{\boldsymbol{U}_i^t\}_{i=1}^l, \mathcal{V}^t = \{\boldsymbol{V}_i^t\}_{i=1}^l\}$
11:     $\boldsymbol{W}^t \leftarrow$ PerturbParams $(\boldsymbol{W}^t, \mathcal{U}^t, \mathcal{V}^t, r, \varepsilon, s^t)$
12:     $\ell_+^t \leftarrow \mathcal{L}(\boldsymbol{W}^t; \mathcal{B}^t)$
13:     $\boldsymbol{W}^t \leftarrow$ PerturbParams $(\boldsymbol{W}^t, \mathcal{U}^t, \mathcal{V}^t, r, -2\varepsilon, s^t)$
14:     $\ell_-^t \leftarrow \mathcal{L}(\boldsymbol{W}^t; \mathcal{B}^t)$
15:     $\boldsymbol{W}^t \leftarrow$ PerturbParams $(\boldsymbol{W}^t, \mathcal{U}^t, \mathcal{V}^t, r, \varepsilon, s^t)$
16:     $\rho^t \leftarrow \left(\ell_+^t - \ell_-^t\right)/(2\varepsilon)$
17:     ResetGenerator$(s)$ {Reset random number generator with seed $s$ }
18:     **for** $i = 1, 2, \dots, l$ **do**
19:       $\boldsymbol{Z}_i^t \leftarrow$ GeneratePerturbMatrix$(r)$ { Regenerate the perturbation matrix $\boldsymbol{Z}_i^t \in \mathbb{R}^{r \times r}$ whose entries are sampled from $\mathcal{N}(0,1)$}
20:       $\boldsymbol{W}_i^{t+1} \leftarrow \boldsymbol{W}_i^t - \eta^t \rho^t \left(\boldsymbol{U}_i^t \boldsymbol{Z}_i^t \boldsymbol{V}_i^{t\mathsf{T}}\right)$
21:     **end for**
22: **end for**

---

In contrast, low-frequency updates risk not capturing the dynamic variations in the gradient subspace throughout the training process.

Therefore, we propose a periodic subspace update strategy. As presented in lines 4-10 in Algorithm 3, we use the MeZO method to calculate the gradient once at the start step and every $F$ steps thereafter. Then the obtained gradient is used to update the projection matrices $\boldsymbol{U}$ and $\boldsymbol{V}$, and keep them unchanged in the subsequent steps. We have experimentally proved the effectiveness and necessity of this strategy. As shown in Table 3, the appropriate update frequency can not only ensure efficiency but also bring significant improvements to model performance.

Despite reducing computational complexity, this strategy will only cause minimal extra memory usage, as presented in Table 1. We adopt a layer-wise parameter update strategy by updating only the parameters of a certain layer of the model at the same time. This means that during the entire training process, we only need to store two additional small matrices at the same time, including the projection matrices $\boldsymbol{U} \in \mathbb{R}^{m \times r}$ and $\boldsymbol{V} \in \mathbb{R}^{n \times r}$, where r is much

Table 1: Memory cost of methods in fine-tuning RoBERTa-large.

| Method | Total GPU Memory |
|---|---|
| MeZO [29] | 2.042GB |
| LOZO [5] | 2.042GB |
| SVD-0 | 2.562GB |

smaller than the dimension of the parameter matrix $\theta \in \mathbb{R}^{m \times n}$. Therefore, the memory usage introduced by the two matrices remains at the same low level as that introduced in [43]. This strategy makes our method almost consistent with the memory required by the MeZO [29] method without any performance loss, and maintains the memory-saving advantage of the ZO method.

## 5  Convergence Analysis

In this section, we theoretically analyze the convergence of our proposed SVD-0. Following the derivations in [43, 30] and [47], we first introduce our proposition and the corresponding lemma.

**Lemma 1** *(**Low-rank subspace of weight matrices [47]**). Gradient matrices become low-rank during fine-tuning. The weight matrix update can be formed as*

$$\theta_T = \theta_0 + \eta \sum_{t=0}^{T-1} \widetilde{\nabla} f(\boldsymbol{\theta})_t, \quad \widetilde{\nabla} f(\boldsymbol{\theta})_t = U_t(U_t^\top f(\boldsymbol{\theta})_t V_t)V_t^\top, \tag{3}$$

*where $\eta$ is the learning rate, $U_t \in \mathbb{R}^{m \times r}$ and $V_t \in \mathbb{R}^{n \times r}$ are projection matrices and can be approximated by the spectrum of $\nabla f(\boldsymbol{\theta})_t$ through $(U, V) = SVD(\nabla f(\boldsymbol{\theta})_t)$.*

Lemma 1 shows that subspace projection matrices can be approximated by adopting SVD on gradients. Given that the SPSA is an unbiased approximation of the exact gradient $\nabla f(\boldsymbol{\theta})$, we can use the SPSA gradient to compute the two projection matrices.

**Proposition 1** *(**Block-diagonal matrix based on SVD**). The singular matrices $\boldsymbol{U}$ and $\boldsymbol{V}$ are column-orthogonal. Therefore, we can similarly define the following notations based on Equation 1:*

$$\boldsymbol{P} = \mathrm{bdiag}(\boldsymbol{V}_1 \otimes \boldsymbol{U}_1, \dots, \boldsymbol{V}_l \otimes \boldsymbol{U}_l),$$

$$\boldsymbol{z} = \left[ \mathrm{vec}(\boldsymbol{Z}_1)^\top, \dots, \mathrm{vec}(\boldsymbol{Z}_l)^\top \right]^\top, \tilde{\boldsymbol{z}} = \left[ \mathrm{vec}(\tilde{\boldsymbol{Z}}_1)^\top, \dots, \mathrm{vec}(\tilde{\boldsymbol{Z}}_l)^\top \right]^\top.$$

Proposition 1 shows that the projection matrices in our method have the same properties as the column-orthogonal matrices used in [43]. Therefore, the subsequent theoretical analysis can proceed in the same way as that proved in [43].

**Lemma 2** *(**Bounded gradient estimation error [43]**). For the gradient estimation in Equation 2, the following two properties hold.*

*(1) By using gradient estimation in Equation 2, the estimated gradient $\widehat{\nabla} f(\boldsymbol{\theta})$ is equivalent to*

$$\widehat{\nabla} f(\boldsymbol{\theta}) = \frac{f(\boldsymbol{\theta} + \varepsilon \boldsymbol{P} \boldsymbol{z}) - f(\boldsymbol{\theta} - \varepsilon \boldsymbol{P} \boldsymbol{z})}{2\varepsilon} \boldsymbol{P} \boldsymbol{z}, \tag{4}$$

*where $\boldsymbol{z} \sim \mathcal{N}(\boldsymbol{0}, \boldsymbol{I}_q)$, $\varepsilon > 0$, $\boldsymbol{P} \in \mathbb{R}^{d \times q}$ satisfies $\boldsymbol{P}^\top \boldsymbol{P} = \boldsymbol{I}_q$ with $d = \sum_{i=1}^l m_i n_i$ and $q = lr^2$.*

*(2) Let $\boldsymbol{z} \sim \mathcal{N}(\boldsymbol{0}, \boldsymbol{I}_q)$, and $f \in C_{L_2}^{2,2}(\mathbb{R}^d)$. Based on Equation 4 whose properties have been analyzed in [30], our method has the same bounded gradient estimation error as that in [43]:*

$$\left\| \mathbb{E}_{\boldsymbol{z}} \left[ \widehat{\nabla} f(\boldsymbol{\theta}) \right] - \boldsymbol{P} \boldsymbol{P}^\top \nabla f(\boldsymbol{\theta}) \right\|_2 \le \frac{\varepsilon^2}{6} L_2 (q+4)^2. \tag{5}$$

*Note that $f \in C_L^{s,p}(S)$ denotes the class of s-th smooth and p-th L-smooth functions over the set S.*

**Theorem 1** *(**Convergence of SVD-0**). Consider the optimization problem $\boldsymbol{x}^* = \arg\min_{\boldsymbol{x} \in \mathbb{R}^d} f(\boldsymbol{x})$, in which $f \in C_{L_1}^{1,1}(\mathbb{R}^d)$ and $f$ exhibits non-convex behavior. Define the stochastic sequence $\mathcal{E}_k = (\boldsymbol{z}_0, \boldsymbol{z}_1, \dots, \boldsymbol{z}_k)$, where each $\boldsymbol{z}_k$ follows the normal distribution $\mathcal{N}(\boldsymbol{0}, \boldsymbol{I}_q)$. Set the step-size parameter as $\eta = \frac{1}{4(q+4)L_1}$. Let $\{\boldsymbol{x}_k\}_{k>0}$ denote the iterates produced via Algorithm 3. For SVD-0, we establish its convergence rate as:*

$$\frac{1}{T} \sum_{k=0}^{T-1} \mathbb{E}_{\mathcal{E}_k} \left[ \left\| \nabla f(\boldsymbol{x}_k) \right\|^2 \right] \le \varepsilon,$$

*under the scaling $T = \Omega\left(\frac{d}{\varepsilon}\right)$ for $\varepsilon \le \mathcal{O}\left(\frac{1}{q^{3/2} d^{1/2} L_1^{3/2}}\right)$, aligning with prior theoretical derivations.*

Combining Proposition 1 and Lemma 2 within the framework proposed in [43], Theorem 1 proves our SVD-0 achieves a convergence rate of $\mathcal{O}(\frac{d}{T})$, which matches the rate derived in [43].

## 6   Experiments

To evaluate the effectiveness of our approach, we implemented SVD-0 on top of the PyTorch framework (version 20.10). All experiments were carried out on a Linux workstation running CentOS, featuring two NVIDIA A100-40GB GPUs, dual Intel 6240R CPUs, and 384GB of RAM. We designed our experiments to explore the following research questions (RQs).

**RQ1 (Superiority of SVD-0)**: To what extent does SVD-0 outperform SOTA methods in accuracy?

**RQ2 (Impact of Hyperparameters)**: What are the impacts of critical hyperparameters (e.g., learning rate, subspace rank, subspace update frequency) on SVD-0-based fine-tuning?

**RQ3 (Applicability of SVD-0)**: How does SVD-0 perform when fine-tuning models of varying sizes or architectures (e.g., masked or causal language models)?

### 6.1   Experimental Settings

**ZO Baselines.**   Our SVD-0 method was evaluated against six latest ZO optimization algorithms, i.e., MeZO [29], ZO-AdaMU [20], S-MeZO [27], SubZero [43], LOZO [5], and HiZOO [48]. Meanwhile, we examined three memory-efficient inference-only approaches, i.e., zero-shot evaluation, in-context learning (ICL) [4], and linear probing (LP) [22].

**Model Settings.**   In our experiments, we took into account both large-scale autoregressive language models (i.e., OPT-1.3B and OPT-13B [45]) and a masked language model (i.e., RoBERTa-large [26]). In the experiments, all ZO methods used a batch size of 16, except where specified, since larger batches help minimize the gradient approximation variance. We chose MeZO as the main baseline because it is the first widely-adopted ZO optimizer for LLMs, and included the first-order SGD as a reference for optimization. In line with previous research [29, 46], our experiments utilized standardized prompt templates, which are crucial in influencing the performance of ZO methods. Moreover, to ensure a fair comparison, we considered multiple values for each key hyperparameter. For example, we investigated the following hyperparameter configurations for OPT-13B: a learning rate in $\{1e{-}7, 2e{-}7, 5e{-}7, 1e{-}6\}$, $\epsilon = 1e - 3$, a batch size of 16 (except for MultiRC and DROP which have a batch size of 8), a rank in $\{24, 32, 48, 64, 128\}$, and a subspace update frequency in $\{500, 1000, 2000\}$. Please refer to Appendix A for detailed configurations of other models. Similarly to the work in [43], we conducted an exhaustive grid search over hyperparameters for each pairing of ZO methods and LLMs, and used the best results for an equitable comparison.

**Dataset Settings.**   For OPT models, we experimented with the SuperGLUE benchmark [40], which consists of various types of tasks, including classification tasks (e.g., SST-2 [37], RTE [1, 2, 7, 14], CB [8], BoolQ [6], WSC [23], and WIC [33]), multiple choice tasks (e.g., COPA [35] and ReCoRD [44]), and generation tasks (e.g., SQuAD [34] and DROP [11]). Here, for each task, we randomly selected 1000 samples for training, 500 samples for validation, and 1000 samples for testing. For the RoBERTa-large model, in addition to the task SST-2, we investigated three more tasks, i.e., SST-5 [37], SNLI [3], and MNLI [41]. In this case, we fixed the parameter $k$ at 512 throughout the training and validation phases, indicating that 512 samples are allocated for each category. For the testing phase, we randomly chose a total of 1000 samples.

### 6.2   Comparison with State-of-the-Arts (R1)

We compared our proposed SVD-0 method with the SOTA ZO optimizers. The experiments were conducted on the SuperGLUE benchmark employing both the OPT-13B and OPT-1.3B language models of different sizes. Note that in each experiment, we applied the adopted stochastic gradient descent (SGD) or ZO method to all model parameters.

Table 2 compares the fine-tuning performance on SuperGLUE benchmark tasks using the OPT-13B model. Here, we considered three types of fine-tuning methods: i) the traditional fine-tuning method (i.e., SGD with backpropagation; ii) inference-only methods (i.e., Zero-shot, ICL and LP) without fine-tuning; and iii) memory-efficient ZO-based methods. To enable a fair comparison between ZO-based methods, we used the MeZO method here as a reference. We evaluated the overall performance across each classification task category and denoted the improvement in performance compared to the baseline (i.e., MeZO) in the sub-column labeled "Total". For example, the total performance on multiple choice tasks with MeZO and SVD-0 is 169.0 and 171.2, respectively. In this case,

Table 2: Comparison of OPT-13B fine-tuning performance (%) on SuperGLUE, where the best results are presented in **bold** and the second-best results are highlighted with underlines.

| Method | Classification Task | | | | | | | | Multiple Choice Task | | | Generation Task | | | All Task |
|---|---|---|---|---|---|---|---|---|---|---|---|---|---|---|---|
| | SST-2 | RTE | CB | BoolQ | WSC | WIC | MultiRC | Total | COPA | ReCoRD | Total | SQuAD | DROP | Total | Total |
| SGD | 94.9 | 82.3 | 85.7 | 78.4 | 65.3 | 65.8 | 74.2 | - | 90.0 | 82.4 | - | 88.0 | 35.5 | - | - |
| Zero-shot | 58.8 | 59.6 | 46.4 | 59.0 | 38.5 | 55.0 | 46.9 | - | 80.0 | 81.2 | - | 46.2 | 14.6 | - | - |
| ICL [4] | 87.0 | 62.1 | 57.1 | 66.9 | 39.4 | 50.5 | 53.1 | - | 87.0 | 82.5 | - | 75.9 | 29.6 | - | - |
| LP [22] | 93.4 | 68.6 | 67.9 | 59.3 | 63.5 | 60.2 | 63.5 | - | 55.0 | 27.1 | - | 3.7 | 11.1 | - | - |
| MeZO [29] | 92.1 | 71.5 | 71.4 | 74.4 | 61.5 | 60.0 | 60.1 | 0% | 87.0 | 82.0 | 0% | 84.2 | 31.2 | 0% | 0% |
| ZO-AdaMU [20] | 92.1 | 72.9 | 67.9 | 73.0 | 61.5 | 60.7 | 63.0 | 0.02% | **89.0** | **83.0** | **1.78%** | 82.4 | **32.0** | -0.87% | 0.27% |
| S-MeZO [27] | 92.3 | **76.9** | **75.0** | **76.5** | 61.1 | 58.2 | **63.3** | 2.51% | 87.0 | 71.2 | -6.39% | 77.9 | 31.9 | -4.85% | -0.53% |
| HiZOO [48] | 91.3 | 69.3 | 69.4 | 67.3 | 63.5 | 59.4 | 55.5 | -3.12% | 88.0 | 81.4 | 0.24% | 81.9 | 31.3 | -1.91% | -2.21% |
| LOZO [5] | 91.7 | 70.4 | 69.6 | 71.9 | 63.5 | 60.8 | 63.0 | -0.02% | **89.0** | 81.3 | 0.77% | 84.9 | 30.7 | 0.17% | 0.18% |
| SubZero [43] | 92.1 | 74.0 | 73.2 | 75.3 | **65.4** | 60.8 | 61.0 | 2.20% | 88.0 | 82.3 | 0.77% | 84.5 | **32.0** | **0.95%** | 1.70% |
| SVD-0 | **93.6** | 75.5 | 71.4 | 75.2 | 63.5 | **65.4** | 60.6 | **2.89%** | **89.0** | 82.2 | 1.30% | **85.1** | 30.9 | 0.52% | **2.19%** |

SVD-0 improves inference performance by 1.30% compared to MeZO. From the results provided in the "Total" sub-columns, we can find that SVD-0 can always achieve top-2 inference performance. Furthermore, we used the final column to show the relative performance improvement for all tasks. From this column, we can find that SVD-0 achieves the best overall performance. Interestingly, while S-MeZO matches SVD-0 in the number of tasks where it excels, its overall performance, shown in the final column, is noticeably inferior to SVD-0 and even falls short of the reference (i.e., MeZO).

## 6.3 Impacts of Hyperparameters (R2)

Hyperparameters play an important role in fine-tuning. In this experiment, we investigate three key hyperparameters (i.e., subspace update frequency, rank, and learning rate) to evaluate their impacts on fine-tuning performance.

Table 3: Impact of subspace update frequency, where the best results are highlighted in **bold**.

| Frequency | SST-2 | RTE | CB | BoolQ | WSC | WIC | MultiRC | COPA | ReCoRD | SQuAD | DROP |
|---|---|---|---|---|---|---|---|---|---|---|---|
| 50 | 90.5 | 57.0 | 64.3 | 65.0 | 63.5 | 55.6 | 57.5 | 72.0 | **72.4** | 74.2 | 23.0 |
| 500 | 89.5 | 55.6 | 69.6 | 64.1 | **63.5** | 53.9 | 58.1 | **73.0** | 72.2 | **74.3** | 22.9 |
| 1000 | **90.6** | **58.5** | 71.4 | 65.2 | **63.5** | 56.4 | **58.2** | **73.0** | 72.1 | 73.7 | **24.0** |
| 2000 | 89.2 | 56.7 | **73.2** | 64.5 | 62.5 | 57.4 | 58.1 | **73.0** | 71.7 | 72.6 | 23.8 |
| 20000 | 89.8 | 56.3 | 71.4 | **65.3** | 62.5 | **57.5** | **58.2** | 72.0 | 72.1 | 72.6 | 22.6 |

For the subspace update frequency $F$, our aim is to assess the impact of altering the subspace update frequency on model performance in various tasks. We conducted experiments based on SVD-0 and the OPT-1.3B model with a fixed rank of $r = 24$ and a learning rate of $1 \times 10^{-7}$. In this analysis, we evaluated five frequencies at varying magnitudes, specifically selected from the set $\{50, 500, 1000, 2000, 20000\}$. Table 3 provides the experimental results. From this table, we can find that when the frequency is set to 1000 (i.e., the subspace is updated in every 1000 steps), SVD-0 achieves the best performance in six of the eleven tasks. Note that SVD-0-based fine-tuning is not sensitive to the hyperparameter $F$. Therefore, we suggest setting $F$ to 1000 by default for fine-tuning.

We also investigated the rank of hyperspace (i.e., $r$) and the learning rate together. Table 4 presents the fine-tuning performance under various combinations of these two hyperparameters, where the rank is selected from $\{2, 24, 48, 64, 128\}$ and the learning rate is selected from $\{1e-7, 5e-7, 1e-6\}$. All the experimental results are collected based on the SST-2 task using the OPT-1.3B model, with a fixed subspace update frequency of 1000. From this table, we can find that the fine-tuning performance is weak when the rank is low (i.e., $r = 2$). While elevating the rank can enhance fine-tuning performance, once the rank surpasses 24, the extent of this enhancement becomes negligible. Note that at low ranks, the performance can vary significantly with different learning rates. In contrast, increasing rank tends to reduce this variability in performance. Moreover, we can observe a similar trend for the learning rate hyperparameter, where setting the learning rate to $5e-7$ can achieve the best performance for most rank settings. However, when learning rates are increased, the inference performance may worsen.

Table 4: Impacts of rank and learning rate.

| Rank \ LR | 1e−7 | 5e−7 | 1e−6 |
|---|---|---|---|
| 2 | 87.7 | 91.2 | 86.7 |
| 24 | 90.6 | 92.2 | 90.3 |
| 48 | 89.5 | 91.6 | 90.1 |
| 64 | 89.9 | 90.4 | 91.6 |
| 128 | 90.0 | 91.3 | 90.6 |

### 6.4 Impact of Model Sizes and Architectures (R3)

In Table 2, we have evaluated the adaptability of SVD-0 to large-scale LLMs. To further validate the generalizability of our approach, we extended our evaluation to the OPT-1.3B model based on representative tasks of different types, where SST-2 and WIC are classification tasks, ReCoRD is a multiple choice task, and SQuAD is a generation task. Table 5 presents the results of the comparison between four ZO-based fine-tuning methods, where the last column shows the average fine-tuning performance of the four tasks. From this table, we can find that SVD-0 is also well-suited for fine-tuning on small-scale LLMs. Although LOZO delivers the highest performance in this experiment, the difference in the average fine-tuning performance between SVD-0 and LOZO is minimal (i.e., merely 0.2%). Note that SVD-0 achieves better performance than MeZO, the reference method, while SubZero fails to beat MeZO. Moreover, SVD-0 can always achieve better performance than its counterpart (i.e., SubZero) with an average improvement of 0.7%. All these observations substantiate the efficiency of our method in enhancing subspaces for optimizing LLMs.

Table 5: Fine-tuning performance (%) comparison for OPT-1.3B, where the top-2 results are marked in bold and with underlines, respectively.

| Method | SST-2 | WIC | ReCoRD | SQuAD | AVG. |
|---|---|---|---|---|---|
| MeZO [29] | 91.7 | 61.1 | 72.2 | 77.4 | 75.6 |
| LOZO [5] | **93.2** | **62.4** | 71.9 | **78.1** | **76.4** |
| SubZero [43] | 91.9 | 60.7 | 72.0 | 77.6 | 75.5 |
| SVD-0 (Ours) | 93.0 | 61.1 | **73.0** | 77.6 | 76.2 |

Table 6: Fine-tuning performance (%) comparison for RoBERTa-large, where the top-2 results are marked in bold and with underlines, respectively.

| Method | SST-2 | SST-5 | SNLI | MNLI |
|---|---|---|---|---|
| Zero-shot | 79.0 | 35.5 | 50.2 | 48.8 |
| MeZO [29] | 93.7 (0.4) | 53.9 (1.9) | 84.8 (1.1) | 76.6 (0.8) |
| LOZO [5] | 94.1 (0.7) | 53.0 (0.4) | **85.4 (0.8)** | **80.4 (1.0)** |
| SVD-0 (Ours) | **94.4 (0.7)** | **54.4 (0.7)** | 85.4 (1.3) | 80.4 (1.5) |

We investigated the fine-tuning performance of different optimization methods on RoBERTa-large, where we considered four downstream tasks, including two sentiment classification tasks (i.e., SST-2 and SST-5) and two natural language inference tasks (i.e., SNLI and MNLI). For a fair comparison, like the work in [5], we performed fine-tuning on each task five times using different random seeds. Table 6 presents the experimental results, reflecting both the average inference performance and its standard deviation (indicated in parentheses) for each combination of fine-tuning methods and tasks. From this table, we can find that SVD-0 has the best performance compared with SOTA ZO optimization methods, showing the adaptability of our approach to different model architectures.

### 6.5 Discussion

**Limitations.** While the SVD-0 technique improves the ZO subspace fine-tuning approach, the accuracy of the subspace projection matrices is significantly influenced by the precision of the ZO gradients. In smaller models like the OPT-1.3B, the ZO gradients may have a greater approximation error, which can result in decreased precision in obtaining the projection matrices.

**Border Impact.** In this paper, we introduced a new approach to derive more precise projection matrices, which can be used to improve the effectiveness of ZO subspace fine-tuning techniques for LLMs. Our method utilizes SVD on ZO gradients to extract projection matrices, eliminating the need for the memory-demanding FO gradients. Our theoretical convergence analysis in conjunction with the experimental findings demonstrates that our research contributes positively to the advancement of memory-efficient fine-tuning methods for LLMs.

## 7 Conclusion

Although various zeroth-order (ZO) optimization methods have been proposed to enable memory-efficient fine-tuning for large language models (LLMs), due to the use of random subspaces, most of them suffer from inaccurate gradient estimation, resulting in inferior training performance. To address this problem, this paper presents a novel ZO subspace fine-tuning method named SVD-0. By precisely capturing fine-tuning subspaces, SVD-0 enables the construction of projection matrices with higher accuracy to achieve more accurate gradient estimation, thus improving the LLM fine-tuning performance. Extensive experimental findings demonstrate the efficacy of SVD-0 in dealing with complex language modeling tasks. In the future, we plan to combine our SVD-0 method with various parameter quantization methods to further reduce the memory required by LLM fine-tuning.

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

# A  Detailed Experimental Settings

## A.1  Hyperparameter Settings

This section provides a detailed overview of the hyperparameters employed in our grid search across the experiments, as depicted in Tables 7 and 9. For the OPT model, we carried out 20,000 steps for each method. Both the SGD and ZO methodologies were implemented for an identical number of steps. In the remaining RoBERTa experiments, ZO optimization strategies were applied over 100,000 training steps. For both models, we evaluated the validation loss every 1,000 training steps to determine the optimal model checkpoint. In the S-MeZO strategy, the sparsity rate is set to 0.75.

Table 7: The hyperparameter grids used for OPT-13B experiments.

| Method | Hyperparameters | | | | |
|---|---|---|---|---|---|
| | Batch Size | Learning Rate | $\epsilon$ | Rank | Update Interval |
| SGD | 16 | $\{1e{-}4, 1e{-}3, 5e{-}3\}$ | – | – | – |
| MeZO [29] | 16 | $\{1e{-}7, 2e{-}7, 5e{-}7, 1e{-}6\}$ | $1e{-}3$ | – | – |
| S-MeZO [27] | 16 | $\{1e{-}6, 5e{-}6\}$ | $1e{-}3$ | – | – |
| LOZO [5] | 16 | $\{1e{-}7, 1e{-}6\}$ | $\{1e{-}3, 1e{-}4\}$ | $\{1, 2, 4\}$ | $\{50, 100\}$ |
| SubZero [43] | 16 | $\{1e{-}7, 2e{-}7, 5e{-}7, 1e{-}6\}$ | $1e{-}3$ | $\{32, 64, 128, 256\}$ | $\{500, 1000, 2000\}$ |
| SVD-0 | 16 | $\{1e{-}7, 2e{-}7, 5e{-}7, 1e{-}6\}$ | $1e{-}3$ | $\{24, 32, 48, 64, 128\}$ | $\{500, 1000, 2000\}$ |

Table 8: The hyperparameter grids used for OPT-1.3B experiments.

| Method | Hyperparameters | | | | |
|---|---|---|---|---|---|
| | Batch Size | Learning Rate | $\epsilon$ | Rank | Update Interval |
| MeZO [29] | 16 | $\{1e{-}7, 5e{-}7, 1e{-}6\}$ | $1e{-}3$ | – | – |
| LOZO [5] | 16 | $\{1e{-}7, 1e{-}6\}$ | $\{1e{-}3, 1e{-}4\}$ | $\{1, 2, 4\}$ | $\{50, 100\}$ |
| SubZero [43] | 16 | $\{1e{-}7, 5e{-}7, 1e{-}6\}$ | $1e{-}3$ | $\{24, 48\}$ | 1000 |
| SVD-0 | 16 | $\{1e{-}7, 5e{-}7, 1e{-}6\}$ | $1e{-}3$ | $\{8, 24, 48\}$ | $\{50, 500, 1000\}$ |

For all previously mentioned ZO methods, we utilized a consistent learning rate schedule and set the weight decay to zero. Typically, we chose a batch size of 16 for the OPT-1.3B and OPT-13B models across various tasks. Nonetheless, due to limited GPU resources, we reduced the batch size to 8 for the DROP, MultiRC, and SQuAD evaluations.

Table 9: Hyperparameter Grids for RoBERTa-large Experiments

| Method | Hyperparameters | | | | |
|---|---|---|---|---|---|
| | Batch Size | Learning Rate | $\epsilon$ | Rank | Update Interval |
| MeZO [29] | 64 | $\{1e{-}7, 1e{-}6, 1e{-}5\}$ | $1e{-}3$ | – | – |
| LOZO [5] | 64 | $2e{-}7$ | $1e{-}3$ | $\{4, 8\}$ | $\{50, 100\}$ |
| SVD-0 | 64 | $1e{-}6$ | $1e{-}3$ | $\{8, 16, 24\}$ | 1000 |

