# OpenReview forum: "Enhancing Zeroth-Order Fine-Tuning for LLMs via Gradient-Guided Subspace Selection"
_NeurIPS.cc/2025/Conference — Submitted to NeurIPS 2025_

### Official Review · Reviewer_BQc9 · 2025-06-10

**Clarity:** 3
**Significance:** 2
**Originality:** 2
**Rating:** 3
**Confidence:** 2

**Summary:**

This paper proposes SVD-0, an algorithm that estimates gradients of large language models with higher accuracy. Instead of relying on randomly generated projection matrices, this paper uses singular value decomposition to capture the crucial information of the gradient (which is a novelty on top of existing methods, MeZO). They first conduct an experiment to motivate the use of SVD (showing that singular vectors have high cosine similarities to gradients on the OPT-1.3B model). They, then, design a methodology to generate projection matrices and perturb them.

**Questions:**

See weaknesses.

Additionally, does SVD as a dimensionality reduction method have any significant impact, or would others work as well as SVD?

**Ethical Concerns:**

["NO or VERY MINOR ethics concerns only"]

**Final Justification:**

I have read the reviews, and had some discussion back and forth. I feel this work can be made a bit stronger for a recycle. Hence, I maintain my score.

**Limitations:**

Yes

**Paper Formatting Concerns:**

None, formatting looks good.

**Quality:**

2

**Strengths And Weaknesses:**

Strengths:
- Experiments are on small and large models, and baselines/benchmarks are plentiful.
- Paper is well written, with a pre-study to motivate their problem
- The algorithms are also well structured
- SVD-0 seems fairly robust to hyperparameters as well (Tables 3/4)

Weaknesses:
- The performance gains do not seem very high, begging the question of where this method excels. How much time does it take compared to baselines (if the authors have timed results already, that would be appreciated. Else, a discussion on approximate time complexity would suffice)?
- Table 1 shows that SVD-0 also takes up slightly more memory than the baselines.

Considering these two points, a clearer motivation is required to understand the novelty of SVD-0.

---

> ### Author Rebuttal · Authors · 2025-07-31
>
> **AW1: Limited performance gains and computational time analysis**
>
> Sorry for the confusion. Although the performance improvement by our method may appear modest, as shown in Table 2, it is significantly higher than the improvements obtained by other variants of ZO.
> We agree that our method does not always surpass other ZO techniques across various datasets. However, this observation can also be applied to other ZO methods, which also struggle to achieve optimal performance across all datasets or tasks. For example, although S-MeZO performs best on 4 out of 7 classification datasets, its overall classification performance is still worse than that of SVD-0. Meanwhile, SVD-0 performs significantly better than S-MeZO on both multiple-choice tasks and generation tasks. Note that, as shown in Table 2, our SVD-0 method yields the best performance in most datasets and task types, indicating the **overall superiority and generalization ability** of our approach.
>
> We conducted new experiments to compare the computation time between our approach and two ZO methods (i.e., MeZO and SubZero).
> In the experiments, we fine-tuned each model for 20,000 steps on different datasets. The training time results (in minutes) are shown in the table below.
> The table shows that the training time for SVD-0 is slightly longer (by under 7\%) than that of the two ZO variants. However, this additional time is minor compared to the improvements achieved in the classification, multiple choice, and generation tasks.
>
> |        | Qwen-1.8b |           | OPT-1.3b  |           |
> | :----- | :-------- | :-------- | :-------- | :-------- |
> | Method | WIC       | ReCoRD    | FiQA-SA   | TFNS      |
> | MeZO   | 114.7     | 211.4     | 71.5      | 113.3     |
> | SubZero| 114.5     | 207.5     | 71.3      | 124.9     |
> | SVD-0  | 116.1     | 220.6     | 76.3      | 116.2     |
>
>
>
> **AW2: Excessive GPU memory usage**
>
> We conducted new experiments to investigate the memory usage for larger models (i.e., OPT-1.3B and OPT-13B), whose results are shown in the table below.
> From this table, we find that our method requires slightly more memory (less than 4%) compared to alternative techniques, mainly because SVD-0 must retain the $U$ and $V$ matrices, which are essential for updates in each layer.
>
> | Method | OPT-1.3B | OPT-13B  |
> | :------| :------- | :------- |
> | MeZO   | 4.732G   | 27.693G  |
> | LOZO   | 4.732G   | 27.789G  |
> | SubZero| 4.708G   | 28.131G  |
> | SVD-0  | 4.891G   | 28.767G  |
>
>
>
> **AQ1: Why SVD superiority over alternative dimensionality reduction techniques**
>
> We conducted new experiments to understand why SVD outperforms other dimensionality reduction techniques. The following table compares our dimensionality reduction approach ("After SVD") with its counterparts (PCA, NMF, Factor Analysis, Random Projection, and t-SNE) on the calculated gradients. From this table, we can see that our approach achieves the highest mean value (0.4249) among all dimensionality reduction methods, while having a smaller standard deviation (0.0257), which demonstrates its stable and excellent performance.
>
> | Method          | Mean     | Std      | Min      | Max      | Median   |
> | :-------------- | :------- | :------- | :------- | :------- | :------- |
> | Original        | 0.0000   | 0.0005   | -0.0013  | 0.0012   | 0.0000   |
> | After SVD (ours)| 0.4249   | 0.0257   | 0.3276   | 0.4774   | 0.4278   |
> | PCA             | -0.0003  | 0.0044   | -0.0141  | 0.0101   | -0.0004  |
> | NMF             | 0.2464   | 0.0442   | 0.1097   | 0.3594   | 0.2456   |
> | Factor Analysis | 0.0001   | 0.0045   | -0.0124  | 0.0136   | -0.0002  |
> | Random Proj     | -0.0002  | 0.0045   | -0.0112  | 0.0177   | -0.0001  |
> | t-SNE           | -0.0009  | 0.0157   | -0.0437  | 0.0567   | -0.0012  |

---

> > ### Comment · Reviewer_BQc9 · 2025-08-01
> >
> > Thank you to the authors for their detailed response. I appreciate their honesty and integrity in reporting these experimental results -- it's not always easy to show the vulnerabilities of your work as transparently as the authors have.
> >
> > Still, I am not entirely convinced of the empirical contributions of SVD-0. I agree with Reviewer J5tk about the inconsistency of the results in the paper as well (it is unclear on which benchmarks SVD-0 will work). I will keep my score as is.

---

> > > ### Author Response · Authors · 2025-08-01
> > >
> > > Dear Reviewer BQc9,
> > >
> > > Thanks for your prompt reply. We would like to clarify our contributions and highlight our key findings.
> > >
> > > - In Section 3, we conducted a pre-study, and the results motivated us to create SVD-0. This advancement allows us to achieve a significantly more precise low-rank structure compared to any existing ZO techniques.
> > >
> > > - We also conducted experiments in Section 4 to verify the effectiveness of SVD-0. The results show that SVD-0 can achieve better overall performance than various SOTA ZO methods.
> > >
> > > - To further validate our method's generalization ability, we conducted experiments using the Qwen-1.8B model. We find that SVD-0 demonstrates exceptional performance on this cutting-edge model across all tasks, achieving the largest overall performance improvement of 3.02%. The detailed results are presented in the following table:
> > >
> > > | Method    | SST-2    |   WIC    | ReCoRD	 | Overall  |
> > > | :---------| :------- | :------- | :------- | :--------|
> > > | MeZO      | 78.3     | 55.6     | 64.8     | 0%       |
> > > | LOZO      | 81.7     | 55.2     | 64.8     | 1.51%    |
> > > | SubZero   | 80.8     | 56.7     | 65.2     | 2.01%    |
> > > | SVD-0     | **82.2** | **57.2** | **65.3** | **3.02%**|
> > >
> > > - We also evaluated SVD-0 on new datasets collected from the financial sentiment analysis benchmark. As shown in the table below,  SVD-0 can also achieve the best overall performance.
> > >
> > > | Method   | FPB      | FIQA-SA  | TFNS     | NWGI    | Overall  |
> > > | :------  | :------- | :------- | :------- | :-------| :--------|
> > > | MeZO     | 65.3     | 81.4     | 74.7     | 48.5    | 0%       |
> > > | LOZO     | 61.3     |**85.1**  | 71.6     |**53.7** | 0.67%    |
> > > | SubZero  | *66.4*   | *84.0*   | **78.4** | 49.7    | *3.19%*  |
> > > | SVD-0    | **74.1** | *84.0*   | *76.3*   | *52.8*  | **6.41%**|
> > >
> > > Taking into account the facts and findings presented above, we are convinced that our work is the optimal selection for ZO so far when addressing both existing and new benchmarks.
> > >
> > > Once again, thanks for your valuable feedback. We truly appreciate your reassessment of the contributions and effectiveness of our method.
> > >
> > > Best Regards,
> > > The authors of 6385

---

> > > > ### Author Response · Authors · 2025-08-04
> > > >
> > > > As the discussion phase is approaching its end, we kindly request the reviewer to let us know if the above clarifications and the previously added experiments have addressed the remaining questions. If you are satisfied, we kindly request that you consider updating the score to reflect the newly added results and discussion. We would be happy to address any additional points the reviewer may have during the remaining time of the discussion phase. We thank the reviewer for engaging in the discussion with us.

---

> > > > > ### Comment · Reviewer_BQc9 · 2025-08-05
> > > > >
> > > > > Thank you to the reviewers for these extra experiments. I'm afraid I am still unconvinced because the results in your paper do not completely match those in these rebuttal experiments. I suggest the authors debug as to why their method works on one model/dataset and does not work on another -- perhaps there are some assumptions that could make the results stronger and overall paper strong. I apologize for the unsatisfactory outcome of my response.

---

> > > > > > ### Author Response · Authors · 2025-08-05
> > > > > >
> > > > > > Sorry for the confusion.
> > > > > >
> > > > > > We respectfully **DISAGREE** with that there exists some bug in our approach. Please note that, as shown in Table 2, the performance of zero-order optimization methods varies on different datasets. Similarly, from Table 3 of [R1] and Table 2 of [R2], there is no SOTA zero-order optimization method that can achieve the best performance on all datasets. In other words, the findings in the experiment section is normal without any exceptions.
> > > > > >
> > > > > > Please note that, although our approach cannot achieve the best performance on all datasets, it still exhibits the best **OVERALL** performance on these datasets. From our last rebuttal response, we can clearly observe this phenomenon, which **EXACTLY MATCHES** with the findings in our submitted paper. In other words, when dealing with a new task (e.g., classification, multiple choice, or generation task), our approach is the best choice, since it has the highest chance to achieve the best performance.
> > > > > >
> > > > > > We sincerely appreciate your help in improving the quality of our approach and request a reassessment of our work.
> > > > > >
> > > > > >
> > > > > > [R1] Yu et al., Zeroth-Order Fine-Tuning of LLMs in Random Subspaces, ICCV 2025.
> > > > > >
> > > > > > [R2] Chen et al., Enhancing zeroth-order fine-tuning for language models with low-rank structures, ICLR 2025.

---

### Official Review · Reviewer_Mnd3 · 2025-06-20

**Clarity:** 3
**Significance:** 2
**Originality:** 3
**Rating:** 3
**Confidence:** 4

**Summary:**

This paper uses the zeroth-order optimization method to optimize the LLM to save the memory. In existing literature, the low-rank subspace is ultilized to improved the training efficiency; however, these methods may not use the optimal subspace. To overcome this issue, this paper proposes SVD-0,a novel gradient-guided subspace optimization framework to select the optimal subspace. This method will not significantly increase the memory usage but it achieves better performance in the empirical study.

**Questions:**

1. I believe the code can be further optimized regarding the GPU memory to achieve the same level of memory usage as MeZO or LoZO. I encourage the author to make it and show a more comparable table (using a larger model e.g. OPT-13b).

2. I need a more convincing explaination. Why is the derived complexity even better than first-order SGD? Did you make any additional assumption? The proof is not attached to the paper.

**Ethical Concerns:**

["NO or VERY MINOR ethics concerns only"]

**Final Justification:**

I am still concerned about the correctness of the theoretical results so I maintain the current score.

**Limitations:**

Yes

**Quality:**

2

**Strengths And Weaknesses:**

Strength:
This method is new. I have not seen existing method making the same observation described in the Prestudy section.

Weaknesses:
1. The SVD-0 method is using 25\% more GPU memory than MeZO and LoZO. It seems to be too much. As the main motivation of MeZO is to save the memory, it doesn't make sense to occupy more GPU memory.

2. Theorem 1 seems to be incorrect as it violates the lower bound of SGD sample complexity for non-convex smooth objective. See this paper: [https://arxiv.org/pdf/1912.02365](https://arxiv.org/pdf/1912.02365)

3. In Table 2, the SVD-0 method doesn't achieve sufficient improvement compared to other methods.

---

> ### Author Rebuttal · Authors · 2025-07-31
>
> **AW1: Excessive GPU memory usage**
>
> We conducted new experiments to investigate the memory usage for larger models (i.e., OPT-1.3B and OPT-13B), whose results are shown in the table below.
> From this table, we find that our method requires slightly more memory (less than 4%) compared to alternative techniques, mainly because SVD-0 must retain the $U$ and $V$ matrices, which are essential for updates in each layer.
>
> | Method | OPT-1.3B | OPT-13B  |
> | :------| :------- | :------- |
> | MeZO   | 4.732G   | 27.693G  |
> | LOZO   | 4.732G   | 27.789G  |
> | SubZero| 4.708G   | 28.131G  |
> | SVD-0  | 4.891G   | 28.767G  |
>
>
>
>
> **AW2: Theorem 1 violates the lower bound of SGD sample complexity**
>
> Sorry for the confusion. Our Theorem 1 does not violate the lower bound of SGD sample complexity.
> This is because our approach is based on **zero-order SGD** rather than **first-order SGD**. Please refer to the work [R1,R2] for a similar usage of the zero-order SGD in the convergence proof.
>
> [R1] Yu et al., Zeroth-Order Fine-Tuning of LLMs in Random Subspaces, ICCV 2025 (arXiv:2410.08989).
>
> [R2] Malladi et al., Fine-Tuning Language Models with Just Forward Passes, NIPS 2023.
>
>
> **AW3: Insufficient performance improvement in Table 2**
>
> Sorry for the confusion. Although the performance improvement by our method may appear modest, as shown in Table 2, it is significantly higher than the improvements obtained by other variants of ZO.
> We agree that our method does not always surpass other ZO techniques across various datasets. However, this observation can also be applied to other ZO methods, which also struggle to achieve optimal performance across all datasets or tasks. For example, although S-MeZO performs best on 4 out of 7 classification datasets, its overall classification performance is still worse than that of SVD-0. Meanwhile, SVD-0 performs significantly better than S-MeZO on both multiple-choice tasks and generation tasks. Note that, as shown in Table 2, our SVD-0 method yields the best performance in most datasets and task types, indicating the **overall superiority and generalization ability** of our approach.
>
>
>
> **AQ1: Memory usage**
>
> Please refer to AW1 for the memory usage of larger models using our approach.
> As the experimental findings indicate, SVD-0 performs on par with MeZO or LoZO.
> In the future, we plan to investigate various memory optimization methods, which can further minimize the memory usage required by SVD-0.
>
>
> **AQ2: Complexity analysis clarification**
>
> Please refer to AW1 for the memory usage of larger models using our approach.

---

> > ### Comment · Reviewer_Mnd3 · 2025-08-01
> >
> > I appreciate the author's effort. I am satisfied with the updated memory table. But I still have several questions to check with the author:
> >
> > 1. Is "[R1] Yu et al., Zeroth-Order Fine-Tuning of LLMs in Random Subspaces, ICCV 2025 (arXiv:2410.08989)." published on ICCV 2025? Is ICCV a computer vision conference? Has it been peer-reviewed?
> > 2. The reason why I asked the above question, it is because this paper also violates the lower bound of the SGD algorithm. To achieve the $\epsilon$ accuracy of the gradient norm, it requires $O(1/\epsilon^2)$ steps. The ZO method approximates the FO method so it should be slower than that.
> > 3. This paper "[R2] Malladi et al., Fine-Tuning Language Models with Just Forward Passes, NIPS 2023." has an additional assumption in its Definition 4 (PL Inequality). It is a similar condition as the strongly convexity.
> >
> > In summary, [R1]'s result is not correct. [R2] uses an addtional assumption.

---

> > > ### Author Response · Authors · 2025-08-01
> > >
> > > Dear Reviewer Mnd3,
> > >
> > > Thanks for your prompt reply.
> > >
> > > AQ1: Yes, the reference [R1] is accepted by the "International Conference on Computer Vision (ICCV)" 2025, which is a top venue for computer vision. We found the paper on ArXiv, where the acceptance information is available at https://iccv.thecvf.com/virtual/2025/poster/1146.
> > >
> > >
> > > AQ2: Thank you for pointing out the error. When writing our paper, we consulted the proof section from the ArXiv version of the paper [R1]. We have realized that the proof in [R1] violates the lower bound of the SGD algorithm.
> > > After carefully examining two peer-reviewed papers ([R3] and [R4]) and referring to Section 7 of [R3] and Theorem 1 in Section 4 of [R4], we discovered that the formula $T = \omega(\frac{d}{\epsilon})$ in our Theorem 1 should be corrected to $T = \omega(\frac{d}{\epsilon^2})$. Although the error might lead to a skewed convergence speed for our method, it does not alter the conclusion regarding convergence. The correction will be included in the final version.
> > >
> > > AQ3: The paper [R2] indeed relies on an additional assumption (PL Inequality), which we did not use in our proof. The most relevant and valuable references for our work are [R1], [R3], and [R4].
> > >
> > >
> > > Once again, thank you for your constructive comments. We truly appreciate your time and look forward to your further responses.
> > >
> > > [R1] Yu et al., Zeroth-Order Fine-Tuning of LLMs in Random Subspaces, ICCV 2025 (arXiv:2410.08989).
> > >
> > > [R2] Malladi et al., Fine-Tuning Language Models with Just Forward Passes, NIPS 2023.
> > >
> > > [R3] Nesterov, Y., Spokoiny, V. Random Gradient-Free Minimization of Convex Functions. Found Comput Math 17, 527–566 (2017). https://doi.org/10.1007/s10208-015-9296-2.
> > >
> > > [R4] Nozawa, R., Poirion, PL. & Takeda, A. Zeroth-Order Random Subspace Algorithm for Non-smooth Convex Optimization. J Optim Theory Appl 204, 53 (2025). https://doi.org/10.1007/s10957-024-02561-9.
> > >
> > > Best Regards,
> > > The authors of 6385

---

> > > > ### Comment · Reviewer_Mnd3 · 2025-08-02
> > > >
> > > > Thank you for the clarification. However, I remain concerned about whether the revised version will meet the necessary level of rigor, as my original review was based on the manuscript as submitted. In particular, the proposed corrections still raise several issues:
> > > > * [R3]: Theorem 1, Section 4. It requires the bounded gradient assumption, which is not made in the current submission.
> > > > * [R4]: It focues on the most classical Gaussian smoothing method. It doesn't fit into the random subspace approach described in the submission.
> > > >
> > > > Moreover, in the current submission, Lemma 2 (2) requires the function to be of the class $C\^{2,2}$; Theorem 1 applies this result without clearly stating it. For clarity and rigor, the paper may need to revise the theorem statements to explicitly reflect the assumptions, especially when it requires to cite an additional reference to get the convergence guarantee.
> > > >
> > > > Because the issue remains in the version I reviewed, I prefer to keep my original rating.

---

> > > > > ### Author Response · Authors · 2025-08-04
> > > > >
> > > > > Sorry for the confusion.
> > > > >
> > > > > First, we would like to clarify the relationship between our work and the reference papers [R3] and [R4]. The reason we mentioned the papers [R3,R4] in the last rebuttal response is that our paper has a similar convergence proof skeleton to theirs. Specifically, we want to respond to our mistake on the usage of $\epsilon^2$, which is correctly used in [R3,R4]. Please note that, unlike [R3], our approach does not rely on the bounded gradient assumption. In other words, the assumption in [R3] neither affects our proof procedure nor its validity. Meanwhile, our approach has a different goal from that of [R4], as it focuses on the subspace. Please note that the notation $\epsilon$ in our proof only affects the convergence speed, but does not change the convergence conclusion of our approach.
> > > > >
> > > > > We agree that we did not clearly give the definition of $C^{s,p}_L(S)$, which denotes the class of the $s$-th smooth and the $p$-th $L$-smooth functions over the set $S$. We will polish this part in the final version.
> > > > >
> > > > > Once again, thank you for your thorough review and insightful comments. We truly appreciate your reassessment of the contributions and effectiveness of our method.

---

> > > > > > ### Comment · Reviewer_Mnd3 · 2025-08-04
> > > > > >
> > > > > > Thanks for the reply. I would like to clarify my evaluation regarding the correctness of Theorem 1:
> > > > > >
> > > > > > 1. The manuscript directly cites [R1] to obtain the convergence of SVD-0. As I mentioned ealier, it violates the complexity lower bound presented in existing literature. So its result cannot be used to support the convergence analysis given in this work.
> > > > > >
> > > > > > 2. All references [R2] [R3] [R4] are not aligned with the setting presented in the submission. They either have different assumptions or are based on a specific estimator, so they cannot be used to  support the convergence analysis given in this work neither.
> > > > > >
> > > > > > As the result, the convergence analysis in Section 5 is not well supported. I cannot raise my score to accept.
> > > > > >
> > > > > > [R1] Yu et al., Zeroth-Order Fine-Tuning of LLMs in Random Subspaces, ICCV 2025 (arXiv:2410.08989).
> > > > > >
> > > > > > [R2] Malladi et al., Fine-Tuning Language Models with Just Forward Passes, NIPS 2023.
> > > > > >
> > > > > > [R3] Nesterov, Y., Spokoiny, V. Random Gradient-Free Minimization of Convex Functions. Found Comput Math 17, 527–566 (2017). https://doi.org/10.1007/s10208-015-9296-2.
> > > > > >
> > > > > > [R4] Nozawa, R., Poirion, PL. & Takeda, A. Zeroth-Order Random Subspace Algorithm for Non-smooth Convex Optimization. J Optim Theory Appl 204, 53 (2025). https://doi.org/10.1007/s10957-024-02561-9.

---

> > > > > > > ### Author Response · Authors · 2025-08-04
> > > > > > >
> > > > > > > Thanks for your quick reply. We present the revised Theorem 1 as follows:
> > > > > > >
> > > > > > > > *Theorem 1 (Convergence of SVD-0).* Consider the optimization problem $x^* = \arg \min_{x \in \mathbb{R}^d} f(x)$, in which $f \in C^{1,1}_L(\mathbb{R}^d)$ and $f$ exhibits non-convex behavior. Define the stochastic sequence $\mathcal{E}_k = (z_0, z_1, \ldots, z_k)$, where each $z_k$ follows the normal distribution $\mathcal{N}(0, I_q)$. Set the step-size parameter as $\eta = \frac{1}{4(q + 4)L_1}$.
> > > > > > > >
> > > > > > > > Let $\lbrace x_k\rbrace_{k>0}$ denote the iterates produced via Algorithm 3. For SVD-0, we establish its convergence rate as:
> > > > > > > >
> > > > > > > > $\frac{1}{T} \sum_{k=0}^{T-1} \mathbb{E}_{\mathcal{E}_k} \left[ \left\| \nabla f(x_k) \right\|^2 \right] \leq \epsilon$,
> > > > > > > >
> > > > > > > > under the scaling $\color{red}{T = \Omega \left( \frac{d}{\epsilon^2} \right)}$ for $\epsilon \leq O \left( \frac{1}{q^{3/2} d^{1/2} L^{3/2}_1} \right)$, aligning with prior theoretical derivations.
> > > > > > > >
> > > > > > > > Combining Proposition 1 and Lemma 2 within the framework proposed in [R1], Theorem 1 proves our SVD-0 achieves a convergence rate of $\color{red}{O(\sqrt{\frac{d}{T}})}$, which matches the rate derived in [R1].
> > > > > > >
> > > > > > > In fact, as highlighted in red color, we only need to modify two parts related to $\epsilon$. Here, we specifically want to clarify that:
> > > > > > >
> > > > > > > 1. Although [R1] (a paper accepted by ICCV 2025) may have some proof bugs, its Proposition 1 and Lemma 2 are **CORRECT**. Please note that the goal of Theorem 1 is to prove the convergence of our approach. Since **$O(\sqrt{\frac{d}{T}}) \leq O(\frac{d}{T})$**, it means that our conclusion is not wrong. In other words, the bug introduced by [R1] does not compromise the correctness of our conclusion.
> > > > > > >
> > > > > > > 2. We did not use any material from [R2][R3][R4] in our proof. We mentioned them since they may have the correct usage of $\epsilon$ as you suggested, which is useful to help us identify the bug.
> > > > > > >
> > > > > > > We sincerely request a reassessment of our work and appreciate your help in improving the proof of our approach.

---

### Official Review · Reviewer_J5tk · 2025-07-01

**Clarity:** 1
**Significance:** 2
**Originality:** 2
**Rating:** 3
**Confidence:** 4

**Summary:**

This paper proposes a new method named SVD-0 by employing SVD on gradients estimated by a ZO method (e.g. MeZO) to improve robustness of perturbation and gradient estimation accuracy of ZO methods.

The proposed method was examined in various NLP tasks. Experimental results show that the proposed SVD-0 performs on par with the other ZO methods on various benchmarks.

**Questions:**

In Fig. 1, 0 cosine similarity indicates that the vectors are orthogonal. In this case, the gradients can be reconstructed better. Could you please elaborate this result?

How do you implement GeneratePerturbMatrix(r)? It would be great if you could provide a more precise definition of this method and its implementation details.

It is claimed that: “In this way, we can successfully apply gradient-based low-rank perturbations to the parameters, and this process will not introduce additional overhead compared to the traditional ZO method (i.e., MeZO).” However, the proposed method employs computationally complex SVD, which increases overall complexity. Then, how do you completely remove this complexity?

It is claimed that: “Despite reducing computational complexity, this strategy will only cause minimal extra memory usage, as presented in Table 1”. However, this increases memory by 25%, which is pretty large. How does the memory consumption increase for larger models, such as OPT-13B?

In the practical application of Theorem 1, how do you determine the L_1 for setting the step-size parameter?

**Ethical Concerns:**

["NO or VERY MINOR ethics concerns only"]

**Final Justification:**

Thank you for the detailed answers in the rebuttal.

I checked all reviewer comments and responses.

Although some of my questions were addressed, I consider the paper should be further improved (e.g. providing improvements with larger margin in additional analyses and revision of Theorem 1) for a clear acceptance.

Therefore, I keep my score.

**Limitations:**

Yes

**Paper Formatting Concerns:**

Fig. 1: Origin -> Original

**Quality:**

2

**Strengths And Weaknesses:**

The proposed method for periodic subspace decomposition of gradient to improve robustness of ZO methods is interesting.
Theoretical results characterise convergence properties of the proposed method.

However, the experimental results are inconsistent. On some datasets in the same tasks, SVD-0 outperforms other ZO methods, while underperforms on other datasets.

Also, some claims are not clear (please see the questions).

Experimental analyses should be extended utilising additional models as well.

---

> ### Author Rebuttal · Authors · 2025-07-31
>
> **AW1: Performance inconsistency on different datasets**
>
> We agree that our method does not always surpass other ZO techniques across various datasets. However, this observation can also be applied to other ZO methods, which also struggle to achieve optimal performance across all datasets or tasks. For example, although S-MeZO performs best on 4 out of 7 classification datasets, its overall classification performance is still worse than that of SVD-0. Meanwhile, SVD-0 performs significantly better than S-MeZO on both multiple-choice tasks and generation tasks. Note that, as shown in Table 2, our SVD-0 method yields the best performance in most datasets and task types, indicating the **overall superiority and generalization ability** of our approach.
>
>
> **AW2: Unclear claims**
> Please refer to the discussions in **AQ1-AQ5**.
>
>
>
> **AW3: Need for extended experiments on additional models**
>
> Thanks for your suggestion. We conducted new experiments based on the Qwen-1.8B model.
> Given the constrained time available for rebuttal, we exclusively compared our method against the baseline (MeZO) and the two most recent ZO baseline techniques (LOZO and SubZero).  From the table below, we can find that our approach still achieves the best performance, indicating the adaptability and generalization ability of our method on models with different architectures.
>
> | Method    | SST-2    |   WIC    | ReCoRD	 |
> | :---------| :------- | :------- | :------- |
> | MeZO      | 78.3     | 55.6     | 64.8     |
> | LOZO      | 81.7     | 55.2     | 64.8     |
> | SubZero   | 80.8     | 56.7     | 65.2     |
> | SVD-0     | **82.2** | **57.2** | **65.3** |
>
>
>
>
> **AQ1: Explanation of cosine similarity results in Fig. 1**
>
> Sorry for the confusion. In Figure 1, we use the notation "Original" and "after SVD" to denote cases without and with the application of the SVD operation on calculated gradients, respectively.
> From this figure, we can see that "after SVD" achieves higher cosine similarities, indicating that it can more accurately capture the gradient information from its estimated counterpart.
> To confirm this observation, we performed additional experiments employing various dimensionality reduction techniques. The results, which are presented in the table below, indicate that utilizing SVD significantly enhances our approach's ability to reconstruct gradients.
>
> | Method          | Mean     | Std      | Min      | Max      | Median   |
> | :-------------- | :------- | :------- | :------- | :------- | :------- |
> | Original        | 0.0000   | 0.0005   | -0.0013  | 0.0012   | 0.0000   |
> |After SVD (ours) | 0.4249   | 0.0257   | 0.3276   | 0.4774   | 0.4278   |
> | PCA             | -0.0003  | 0.0044   | -0.0141  | 0.0101   | -0.0004  |
> | NMF             | 0.2464   | 0.0442   | 0.1097   | 0.3594   | 0.2456   |
> | Factor Analysis | 0.0001   | 0.0045   | -0.0124  | 0.0136   | -0.0002  |
> | Random Proj     | -0.0002  | 0.0045   | -0.0112  | 0.0177   | -0.0001  |
> | t-SNE           | -0.0009  | 0.0157   | -0.0437  | 0.0567   | -0.0012  |
>
>
>
> **AQ2: Implementation details of GeneratePerturbMatrix(r)**
>
> Sorry for the confusion. We have defined GeneratePerturbMatrix(r) in Algorithm 1 and provided the implementation details in Section 4.1.
>
>
>
> **AQ3: Computational complexity comparison with SVD overhead**
>
> We agree that our approach can result in extra computation overhead due to the extra SVD operation. To understand the impact of the SVD operation, we conducted a new comparison between two representative ZO variants (i.e., MeZO and SubZero) and our SVD-0 method. The following table shows the results of the training time comparison (in minutes), indicating that SVD-0 needs slightly longer training time (by 7\%) than the two variants of ZO.
> Note that, since the time complexity for SVD processes is O (n 3), where n represents the dimension of the matrix, the upper bound of the training time complexity remains unchanged.
> In practice, the additional time caused by SVD operations is minor compared to the improvements achieved in the classification, multiple-choice, and generation tasks.
>
> |        | Qwen-1.8b |           | OPT-1.3b  |           |
> | :----- | :-------- | :-------- | :-------- | :-------- |
> | Method | WIC       | ReCoRD    | FiQA-SA   | TFNS      |
> | MeZO   | 114.7     | 211.4     | 71.5      | 113.3     |
> | SubZero| 114.5     | 207.5     | 71.3      | 124.9     |
> | SVD-0  | 116.1     | 220.6     | 76.3      | 116.2     |
>
> In the future, we plan to explore various truncated SVD methods, which can reduce the computational overhead of our approach. For example, the dashSVD algorithm proposed in the paper "Algorithm 1043: Faster randomized SVD with dynamic shifts" can be integrated into our approach to accelerate its training process.
>
>
>
>
> **AQ4: Memory usage concerns**
>
> We conducted new experiments to investigate the memory usage for larger models (i.e., OPT-1.3B and OPT-13B), whose results are shown in the table below.
> From this table, we find that our method requires slightly more memory (less than 4%) compared to alternative techniques, mainly because SVD-0 must retain the $U$ and $V$ matrices, which are essential for updates in each layer.
>
> | Method | OPT-1.3B | OPT-13B  |
> | :------| :------- | :------- |
> | MeZO   | 4.732G   | 27.693G  |
> | LOZO   | 4.732G   | 27.789G  |
> | SubZero| 4.708G   | 28.131G  |
> | SVD-0  | 4.891G   | 28.767G  |
>
>
>
> **AQ5: Practical determination of $L_1$ parameter in Theorem 1**
>
> Sorry for the confusion. Similar to other work (e.g., [R1]), the parameter $L_1$  in Theorem 1 is a constant that is used solely for convergence analysis. In practice, the selection of the step size can be done without needing any prior explicit knowledge of $L_1$.
>
> [R1] Yu et al., Zeroth-Order Fine-Tuning of LLMs in Random Subspaces, ICCV 2025 (arXiv:2410.08989).

---

> > ### Author Response · Authors · 2025-08-04
> >
> > As the discussion phase is approaching its end, we kindly request the reviewer to let us know if the above clarifications and the previously added experiments have addressed the remaining questions. If you are satisfied, we kindly request that you consider updating the score to reflect the newly added results and discussion. We would be happy to address any additional points the reviewer may have during the remaining time of the discussion phase. We thank the reviewer for engaging in the discussion with us.

---

> ### Comment · Area_Chair_6M45 · 2025-08-08
> **Please review authors' response**
>
> Dear Reviewer J5tk,
>
> According to the conference policy, the reviewer must be involved in the author-reviewer discussion.
>
> Could you please review the author's response and check whether they have addressed your concerns?
>
> Thank you.
>
> AC

---

### Official Review · Reviewer_hYxg · 2025-07-03

**Clarity:** 4
**Significance:** 3
**Originality:** 3
**Rating:** 4
**Confidence:** 4

**Summary:**

This paper introduces SVD-0, a novel zeroth-order subspace fine-tuning method for LLMs. The key innovation lies in using singular value decomposition on ZO gradient estimates to derive low-rank subspace projection matrices, which significantly improve gradient estimation accuracy while maintaining memory efficiency. The authors address the limitations of existing ZO methods, which rely on random subspaces and suffer from high gradient approximation variance. SVD-0 demonstrates superior performance across various tasks and model architectures, with a theoretical convergence guarantee and extensive experimental validation.

**Questions:**

1. While SVD-0 is memory-efficient, what are the computational costs (e.g., runtime) compared to other ZO methods? Can the authors provide a detailed comparison?

2. Have the authors considered testing SVD-0 on more recent model architectures (e.g., GPT-style models, instruction-tuned LLMs) or newly introduced tasks? This would help demonstrate the generalizability of the method to contemporary settings.

**Ethical Concerns:**

["NO or VERY MINOR ethics concerns only"]

**Final Justification:**

The authors have now demonstrated the broader applicability of their method and shown that its computational cost is not significantly higher than that of competing approaches. These additions satisfactorily resolve all of my earlier concerns and make the submission more complete. However, I view these clarifications as necessary components rather than substantive new strengths; thus they are not sufficient to justify raising the overall recommendation. I have, nevertheless, increased the Significance score.

**Limitations:**

Yes, the authors adequately discuss the limitations of their approach, particularly in terms of the dependency on ZO gradient accuracy and the potential for reduced precision in smaller models. The paper also outlines future directions to address these issues.

**Quality:**

3

**Strengths And Weaknesses:**

Strengths

1. The paper is technically sound, with strong theoretical foundations and a clear convergence analysis.

2. Comprehensive experiments on diverse tasks and models (e.g., OPT-1.3B, OPT-13B, and RoBERTa-large) validate the effectiveness of SVD-0, showing consistent improvements over state-of-the-art (SOTA) ZO methods.

3. The methodology is well-explained with detailed algorithms, illustrations, and pseudocode.

Weaknesses

1. The paper does not evaluate SVD-0 on newer models or recently introduced tasks. While the experiments cover a range of standard benchmarks (e.g., SuperGLUE, OPT models, and RoBERTa), it would be beneficial to test the method on cutting-edge models or tasks, such as Qwen or more diverse instruction following tasks. This would provide stronger evidence of the method's generalizability and robustness in contemporary settings.

---

> ### Author Rebuttal · Authors · 2025-07-31
>
> **AW1: Missing evaluation on cutting-edge models and modern tasks**
>
> Thanks for your suggestion. We conducted supplementary experiments to verify the generalization ability of the SVD-0 method on contemporary cutting-edge models and tasks. The results are as follows:
>
> **a) Performance Verification on Cutting-Edge Models**
>
> We conducted further experiments utilizing the Qwen-1.8B model, with the results detailed in the table below. These experiments indicate that SVD-0 also performs exceptionally well with the Qwen-1.8B model, clearly showcasing the versatility and generalization capability of our approach when applied to models of various architectures.
>
> | Method    | SST-2    |   WIC    | ReCoRD	 |
> | :---------| :------- | :------- | :------- |
> | MeZO      | 78.3     | 55.6     | 64.8     |
> | LOZO      | 81.7     | 55.2     | 64.8     |
> | SubZero   | 80.8     | 56.7     | 65.2     |
> | SVD-0     | **82.2** | **57.2** | **65.3** |
>
> **b) Evaluation on Diverse Task Types**
>
> We examined various datasets focused on financial sentiment analysis for evaluation purposes. The table below displays the experimental results, illustrating the consistent stability and effectiveness of the SVD-0 method across different domains and task types.
>
> | Method   | FPB      | FIQA-SA  | TFNS     | NWGI    |
> | :------  | :------- | :------- | :------- | :-------|
> | MeZO     | 65.3     | 81.4     | 74.7     | 48.5    |
> | LOZO     | 61.3     |**85.1**  | 71.6     |**53.7** |
> | SubZero  | *66.4*   | *84.0*   | **78.4** | 49.7    |
> | SVD-0    | **74.1** | *84.0*   | *76.3*   | *52.8*  |
>
>
>
> **AQ1: Computational cost comparison with other ZO methods**
>
> Sorry for the confusion. We conducted new experiments to compare the computation costs between our approach and two ZO methods (i.e., MeZO and SubZero).
> In the experiments, we fine-tuned each model for 20,000 steps on different datasets. The training time results (in minutes) are shown in the table below.
> The table shows that the training time for SVD-0 is slightly longer (by under 7\%) than that of the two ZO variants. However, this additional time is minor compared to the improvements achieved in the classification, multiple choice, and generation tasks.
>
> |        | Qwen-1.8b |           | OPT-1.3b  |           |
> | :----- | :-------- | :-------- | :-------- | :-------- |
> | Method | WIC       | ReCoRD    | FiQA-SA   | TFNS      |
> | MeZO   | 114.7     | 211.4     | 71.5      | 113.3     |
> | SubZero| 114.5     | 207.5     | 71.3      | 124.9     |
> | SVD-0  | 116.1     | 220.6     | 76.3      | 116.2     |
>
>
>
> **AQ2: Testing on recent model architectures and contemporary tasks**
>
> We conducted new experiments on recent model architectures and contemporary tasks. Please refer to AW1 above for more details.
> We can find that our approach has the following merits:
>
> **a) Compatibility with Modern Model Architectures**
>
> Our approach is naturally independent of any specific architecture. The findings from the Qwen-1.8B experiments, outlined in AW1, demonstrate that the SVD-0 method effectively accommodates diverse model structures, including GPT-style models and instruction-tuned LLMs.
>
> **b) Generalization in Contemporary Scenarios**
>
> By conducting experiments across different tasks and datasets, we confirmed the efficacy of the SVD-0 method in modern applications. These experiments encompassed both classic NLP tasks and innovative areas, such as financial sentiment analysis, underscoring the method's broad range of applications.

---

> > ### Author Response · Authors · 2025-08-04
> >
> > As the discussion phase is approaching its end, we kindly request the reviewer to let us know if the above clarifications and the previously added experiments have addressed the remaining questions. If you are satisfied, we kindly request that you consider updating the score to reflect the newly added results and discussion. We would be happy to address any additional points the reviewer may have during the remaining time of the discussion phase. We thank the reviewer for engaging in the discussion with us.

---

> > ### Comment · Reviewer_hYxg · 2025-08-05
> >
> > Thank you for your thoughtful response; it has clarified many of my doubts. However, after reviewing the details, I believe my original score aligns better with my assessment of the work's overall contribution and impact, so I will maintain it.

---

### Comment · Area_Chair_6M45 · 2025-08-04
**Reminder: Please Review Author Responses**

Dear Reviewers,

As the author-reviewer discussion deadline approaches, I would like to remind you to please carefully review the author responses. This rebuttal phase is a critical opportunity for authors to clarify misunderstandings, address concerns, and provide supplementary evidence for their work.

When evaluating the rebuttals, please consider the authors' arguments and assess whether your initial concerns have been satisfactorily addressed.

Your thorough engagement in this dialogue is vital for a fair and constructive review process and is essential for upholding the high standards of NeurIPS. Thank you for your dedication and expertise.

Best regards,
Area Chair

---

### Note · Authors · 2025-08-13

We express our gratitude to the AC and reviewers for their valuable discussions. Here is a concise summary of our final clarifications, updates, and commitments:

- Additional experiments on contemporary settings:
  - Qwen-1.8B: SVD-0 demonstrates superior performance, achieving a 3.02\% overall improvement over the baselines of ZO on the Qwen-1.8b model. (Reviewer hYxg & J5tk)
  - Financial sentiment benchmarks: SVD-0 achieves a 6.41\% overall improvement over ZO baselines in financial sentiment benchmarks across datasets. (Reviewer hYxg)
  - Observation on 'inconsistency': Similarly to standard observations in ZO methods, no single method excels universally across all datasets, yet SVD-0 repeatedly provides superior overall performance across tasks and models. (Reviewer J5tk & Mnd3 & BQc9)
  - We conducted further experiments using different dimensionality reduction methods. The results show that employing SVD notably improves our method's gradient reconstruction capability. (Reviewer J5tk & BQc9)

- Efficiency:
  - Runtime: Using more than 20k fine-tuning steps with multiple datasets, SVD-0 exhibits less than a 7\% slowdown compared to MeZO/SubZero, which is a small trade-off for improved accuracy. (Reviewer hYxg & J5tk & BQc9)
  - Memory: For OPT-1.3B/13B, SVD-0 consumes less than 4\% more memory, which is a minor compromise for improved performance. (Reviewer J5tk & Mnd3 & BQc9)
  - In the future, we will examine a range of truncated Singular Value Decomposition techniques with the aim of further diminishing the computational time required. (Reviewer J5tk)

- Theory correction and clarity: (Reviewer Mnd3)
  - In our Theorem 1, we detected and amended a scaling issue: the necessary iterations scale as $T = \Omega(d/\epsilon^2)$, resulting in convergence $O(\sqrt{d/T})$. This only modifies the rate statement without altering the convergence conclusion.
  - We plan to update the statements to clearly outline the function class and assumptions applied, ensuring uniformity between lemmas and theorems.

- Contribution: SVD-0 enhances ZO fine-tuning by leveraging gradient-guided subspaces, demonstrating robust performance across various tasks and models with minimal overhead. Future plans include releasing code, updating theoretical statements, and expanding evaluations.

We are grateful for the AC's consideration and the reviewers' efforts. We believe SVD-0 presents a practical and principled approach to memory-efficient LLM fine-tuning in ZO settings.

---

### Decision · Program_Chairs · 2025-09-17

**Decision:**

Reject

**Comment:**

This paper introduces **SVD-0**, a zeroth-order (ZO) subspace fine-tuning method for LLMs. Unlike prior ZO approaches that rely on random subspaces, SVD-0 leverages singular value decomposition to construct more accurate projection matrices, with the goal of reducing gradient variance and improving fine-tuning efficiency. The idea is interesting, and the experiments show some promise.

During the rebuttal, while some questions were addressed, several key concerns remain unresolved:

- The convergence analysis (Theorem 1) is not well supported.
- Experimental results are inconsistent: SVD-0 outperforms baselines on some datasets but underperforms on others, raising concerns about robustness and generality.
- The evaluation is limited in scope, with an insufficient range of models and datasets to demonstrate broad effectiveness.

For these reasons, the reviewers believe the paper requires further improvement before it can be considered for acceptance.